# Alphavirus-Induced Membrane Rearrangements during Replication, Assembly, and Budding

**DOI:** 10.3390/pathogens10080984

**Published:** 2021-08-04

**Authors:** Zeinab Elmasri, Benjamin L. Nasal, Joyce Jose

**Affiliations:** 1Huck Institutes of the Life Sciences, The Pennsylvania State University, University Park, PA 16802, USA; zxe14@psu.edu; 2Department of Biochemistry & Molecular Biology, Eberly College of Science, The Pennsylvania State University, University Park, PA 16802, USA; bln5145@psu.edu

**Keywords:** *Togaviridae*, alphavirus, spherule, replication, cytopathic vacuole, nucleocapsid core, assembly, filopodia, budding

## Abstract

Alphaviruses are arthropod-borne viruses mainly transmitted by hematophagous insects that cause moderate to fatal disease in humans and other animals. Currently, there are no approved vaccines or antivirals to mitigate alphavirus infections. In this review, we summarize the current knowledge of alphavirus-induced structures and their functions in infected cells. Throughout their lifecycle, alphaviruses induce several structural modifications, including replication spherules, type I and type II cytopathic vacuoles, and filopodial extensions. Type I cytopathic vacuoles are replication-induced structures containing replication spherules that are sites of RNA replication on the endosomal and lysosomal limiting membrane. Type II cytopathic vacuoles are assembly induced structures that originate from the Golgi apparatus. Filopodial extensions are induced at the plasma membrane and are involved in budding and cell-to-cell transport of virions. This review provides an overview of the viral and host factors involved in the biogenesis and function of these virus-induced structures. Understanding virus–host interactions in infected cells will lead to the identification of new targets for antiviral discovery.

## 1. Introduction

Alphaviruses are enveloped, single-stranded positive-sense RNA (+RNA) viruses of the genus *Alphavirus* [1]. Since 2018, *Alphavirus* has been the sole genus of the family *Togaviridae*, after the genus *Rubivirus* was reclassified under a novel family, *Matonaviridae* [2]. The *Alphavirus* genus has over 40 members that are divided into two groups, Old World and New World, depending on their area of origin and the disease they cause [3]. Old World alphaviruses cause fever, rash, and arthritis, and well-studied examples include Sindbis virus (SINV), Chikungunya virus (CHIKV), Semliki Forest virus (SFV), and Ross River virus (RRV). New World alphaviruses may cause encephalitis in their hosts and include Venezuelan equine encephalitis virus (VEEV), Western equine encephalitis virus (WEEV), and Eastern equine encephalitis virus (EEEV) [3]. An exception to encephalitic New World is the Mayaro virus (MAYV), which is arthritogenic [4].

Alphaviruses are zoonotic, transmitted primarily by mosquitoes of the *Culicidae* family [5]. Members of the genus *Aedes*, specifically *A. aegypti* and *A. albopictus*, are of particular interest, due to their ability to cause massive outbreaks of CHIKV in urban areas. Ultimately, humans can become infected either via a direct spillover from enzootic habitats or when amplification by mosquito vectors results in high levels of disease circulation [5]. The global spread of alphaviruses is on the rise, likely a result of exploding mosquito populations combined with viral adaptation and increased global urbanization [6].

Alphavirus particles are small and spherical, with diameters of ~65–70 nm, and contain two distinct layers of T = 4 icosahedral symmetry—a nucleocapsid core (NC) consisting of RNA bound to capsid proteins, and an outer glycoprotein layer with E1/E2 glycoprotein spikes embedded within the host-derived lipid bilayer [1]. The alphavirus RNA genome is approximately 12 kb in length, encompassing two open reading frames (ORFs). The 5′-terminal ORF encodes a polyprotein that is processed to yield non-structural replicase proteins (nsPs) 1, 2, 3, and 4, while the 3′-terminal ORF translated from a subgenomic RNA encodes a polyprotein that is processed to yield structural proteins Cp, E3, E2, 6K, TF, and E1 [7,8]. 

Similar to other +RNA viruses, alphaviruses induce the rearrangement and restructuring of cellular membranes and the cytoskeleton during replication and assembly, forming structures known as replication spherules, cytopathic vacuoles (CPVs), and filopodial extensions. However, the mechanisms by which alphaviruses organize and execute these rearrangements are not completely understood. This review will summarize the alphavirus lifecycle and explore our current understanding of the mechanisms behind alphavirus-induced structural modifications, as well as their importance in the progression of the viral lifecycle. We will address the role of both viral and host factors in the biogenesis of these structures and highlight areas that require further studies.

## 2. Overview of the Alphavirus Lifecycle 

Alphaviruses begin the infection process by engaging with molecules on the surface of host cells. Usually proteinaceous, these molecules can be either attachment factors, which allow viral docking with the target cell, or entry receptors, which facilitate virus internalization [9]. The viral E2 glycoprotein is mainly responsible for interacting with attachment factors and entry receptors. While it was originally thought that only E2 was involved in receptor binding, it is now known that E1 plays a role as well [10,11,12]. Heparan sulfate (HS) is an attachment factor used by several alphaviruses; for example, HS expression has been shown to increase the infectivity of SINV, CHIKV, and EEEV, depending on the degree of sulfation [13,14,15,16,17]. C-type lectins, including DC-SIGN and L-SIGN, and phosphatidylserine (PS) receptors such as the T-cell immunoglobulin and mucin domain (TIM) family, have also been implicated in the attachment process. Cells transfected with either DC-SIGN or L-SIGN showed increases in SINV binding and infectivity, and the ectopic expression of TIM-1 increased RRV uptake and infection in cells [18,19,20,21]. Though these attachment factors are known as of now, many questions remain regarding additional attachment factors and interactions between attachment factors and entry receptors.

Very little is known about alphavirus entry receptors, often thanks to a lack of discernable interactions between putative receptors and purified E2 proteins [22]. However, recent progress has been made on implicating certain cell surface proteins in alphavirus entry. Natural resistance-associated macrophage protein (NRAMP) has been proposed as a potential entry receptor for SINV, as the genetic downregulation of NRAMP2 in mouse fibroblasts, and the post-translational downregulation of NRAMP by iron treatment in insect cells all resulted in a reduced SINV infection [23]. Additionally, matrix remodeling associated protein 8 (Mxra8) has been identified as a receptor for many notable arthritogenic alphaviruses, including CHIKV, RRV, and MAYV. Expressed on the surface of epithelial, mesenchymal, and myeloid cells, the ectopic expression of Mxra8 has been shown to directly enhance alphavirus infection. Mxra8 has also been shown through enzyme-linked immunosorbent assay (ELISA) to interact directly with CHIKV particles, and transfection of viral RNA eliminates the need for Mxra8 expression [24]. Though recent structural analyses have revealed how Mxra8 complexes with alphavirus components [11,12], the means through which these interactions facilitate alphavirus internalization remain elusive.

The alphavirus envelope consists of a host-derived lipid bilayer and an embedded glycoprotein layer made of 80 spikes. Each spike is a trimer of E1/E2 heterodimers, resulting in the incorporation of 240 copies of each glycoprotein into a mature virion [8]. Alphaviruses such as SFV and VEEV have been shown to transiently retain E3 in association with E2 after furin cleavage in the mature virion [25,26]. The E1 and E2 glycoproteins are type I transmembrane proteins containing a single transmembrane segment; E2 has a small cytoplasmic domain (cdE2) that specifically binds to the hydrophobic pocket of the capsid protein (Cp) in the NC [27]. The binding of E2 to attachment factors and entry receptors triggers the internalization of the virus via clathrin-mediated endocytosis and subsequent trafficking to the early endosome, where the clathrin coat is rapidly disassembled [28]. As the internalized virus passes through the endosomal pathway, ATP-dependent proton pumps expose the virus to an increasingly low pH in a stepwise fashion, from a pH of ~6.5–6.0 in the early endosome, to a pH of ~6.0–5.0 in the late endosome, to an eventual pH of ~5.0–4.6 [29,30]. The transition to an acidic pH stimulates the disassociation of E1 from E2 and the reorganization of E1 glycoproteins into activated homotrimers [31,32,33,34,35,36,37,38,39]. Each E1 in the homotrimer displays a previously hidden hydrophobic fusion loop; the three fusion loops form a hairpin-like structure that inserts into the late endosomal membrane, causing a conformational change in the E1/E2 heterodimer, triggering the formation of fusion pores in both the cellular and viral membrane to allow for release of the NC into the cytoplasm [3,40,41]. However, studies on SFV and MAYV showed that the E1/E2 conformational changes can be partially reversed upon reneutralization of the acidic PH [42,43]. Low-pH-induced conformational changes of the glycoprotein spikes occur rapidly once sufficient acidification is reached in order to avoid lysosomal degradation [37].

Once the NC has been released into the cytoplasm, disassembly must occur in order to expose the genome for translation. However, this process is not well understood. A model for NC disassembly must reconcile the fact that newly produced NCs are stable in the cell, yet incoming NCs are not [3]. It is known, however, that the uncoating of the NC occurs within the first minute after fusion, and is reliant on the capsid protein (Cp) interaction with 60S ribosomal RNA [44,45,46]. After uncoating, the genomic RNA (gRNA) is translated to yield two polyproteins. Of these polyproteins, ~10% contain nsPs 1, 2, 3, and 4 (P1234), while 90% contain nsPs 1, 2, and 3 (P123) [47]. After translation, the polyproteins containing nsP4 are processed by nsP2 to yield P123, as full-length P1234 is not capable of RNA synthesis [48]. P123 and nsP4 assemble into an early replication complex that facilitates genome replication. After its assembly, the early replication complex synthesizes negative-strand genomes using the positive-strand genome as a template [49]. Eventually, using the negative-strand and double-stranded RNA (dsRNA) intermediates, the replication complex synthesizes new copies of the +RNA genome [49]. Although structural information regarding most of the nsPs is available, because there is no known structure of the replication complex, the trigger that causes the switch from negative-strand to positive-strand synthesis is poorly understood. However, it is believed that the eventual processing of P123 by nsP2 into individual nsP1, nsP2, and nsP3 irreversibly locks the replication complex into a late replication complex formation that is only capable of synthesizing +RNAs, specifically the gRNA and the subgenomic RNA (sgRNA) [47].

Viral RNA synthesis takes place inside bulb-like structures termed spherules located on the plasma membrane and the membranes of cytopathic vacuole type I (CPV-I) structures [50,51,52]. The replication complex synthesizes sgRNA using an internal subgenomic promoter; these sgRNAs correspond to the last third of the genome that encodes the structural polyprotein [6]. After translation, Cp autoproteolytically cleaves itself from the rest of the structural polyprotein and specifically packages gRNA, leading to the formation of new NCs containing 240 copies of Cp [53,54,55]. Though Cp is capable of packaging any RNA into particles, specific nucleotide sequences within the gRNA have been identified that promote its preferential packaging [56,57,58]. 

The signal sequences within the structural polyprotein mediate their translocation and insertion into the endoplasmic reticulum (ER) membrane, where signal peptidase cleaves the structural polyprotein into pE2, 6K, and E1 [59,60]. PE2 is the precursor protein that is eventually processed into E3 and E2 [6]. PE2 and E1 are then post-translationally modified as they pass through the secretory pathway [61,62,63,64,65]. During trafficking in the secretory pathway, pE2 and E1 form a heterodimer in which the E3 portion of pE2 acts as a protective clamp that prevents the premature exposure of the fusion loop at a low pH [66]. Late in the secretory pathway, furin cleaves pE2 to produce E3 and E2, releasing E1 from its protective clamp and forming the mature E1/E2 heterodimer. The furin cleavage of pE2 is required for host cell entry and fusion [67,68]. Cytopathic vacuoles type II (CPV-II), studded with NCs on their cytoplasmic sides and glycoproteins on their luminal sides, originate from the *trans*-Golgi network and are thought to carry the NCs and glycoprotein spikes to the plasma membrane [69,70]. Alphavirus budding is temperature and pH dependent—it occurs ideally at physiological temperatures and at a neutral to slightly alkaline pH [71,72]. 

Cp-E2 interactions are required for budding [73]; it is also known that the E1/E2 heterodimer formation is required for infectious virus assembly and budding [74,75,76]. Recent studies support a model in which the E1/E2 spikes are arranged in a regular icosahedral lattice on the plasma membrane [77]. Interactions between this lattice via cdE2 and a preformed NC from the cytoplasm initiate virus assembly, and the virions subsequently bud from the plasma membrane [78,79,80]. High-resolution cryo-EM structures of the MAYV Cp support this model, revealing charged amino acid pairs at interfaces between Cps from different hexameric or pentameric units, indicating their likelihood to assemble and orient in the cytoplasm with the help of electrostatic interactions prior to budding [78]. Furthermore, neutralizing the anti-CHIKV antibodies that crosslink viral E1 and E2 glycoproteins on the outer leaflet of the plasma membrane has been shown to prevent the NCs from inducing membrane curvature. As a result, NCs cannot bud and remain trapped at the membrane and in the cytoplasm [81].

In addition to the Cp-E2 and E1/E2 interactions, 6K and TF are important for virus assembly and budding, but the mechanisms by which these proteins promote virus assembly and budding remain incompletely understood [7,82,83,84,85]. TF is a structural protein produced ~10–15% of the time from a ribosomal frameshift during the translation of 6K [7]. TF antagonizes interferon production during early infection, while 6K forms ion channels and functions as a viroporin [86,87,88]. E3, 6K, and TF do not need to be present in a viral particle for it to be infectious [89].

## 3. Replication-Induced Membrane Rearrangements 

+RNA viruses hijack and rearrange cellular membranes into unique structures that support the replication of their genome [90,91]. These structures play a role in concentrating replicase components by spatially confining the replication process into specific compartments. In addition, these structures shield replication intermediates such as dsRNA from host defense mechanisms [92]. The membrane structures assembled during the replication of +RNA viruses are termed replication organelles (ROs) and are segregated into the following two morphotypes: the double-membrane vesicles (DMVs) type and the spherule/invagination type [93,94]. DMVs are formed by hijacking membranes of the secretory pathway and have a diameter of ~100–400 nm. These ROs are found in cells infected with coronaviruses [95,96,97,98,99], the hepatitis C virus (HCV) [100,101], picornaviruses [102,103,104,105], noroviruses [106], and arteriviruses [107,108]. For a detailed review of DMVs, please refer to [94]. Other +RNA viruses such as flaviviruses [109,110,111], alphaviruses [47,112,113], nodaviruses [114], and bromoviruses [115,116], induce the formation of invaginated spherules in host organelles. Replication spherules have a diameter of ~30–90 nm and are connected to the cytosol by a narrow channel that allows the import of nucleotides and the export of RNA molecules. The specific organelle localization of replication spherules depends on the virus inducing them. For instance, flavivirus- and bromovirus-induced spherules are connected to the ER membranes, while the nodavirus-induced ones are found on the outer mitochondrial membrane [93].

In the case of alphaviruses, studies conducted on SFV in the 1960s reported the presence of spherules measuring 50–60 nm in diameter on the plasma membrane and on the limiting membranes of large vacuoles (0.6–2 µm in diameter) termed CPV-Is [50]. Cell fractionation and electron microscopy (EM) autoradiography indicated that CPV-Is might be the site of RNA replication [113]. Two decades later, immunofluorescence and immuno-EM techniques revealed electron-dense material at the neck of the spherules along with the non-structural proteins nsP3 and nsP4 [117]. The same study used endosomal tracers and lysosomal markers to show that CPV-Is originate from endo-lysosomal membranes [117]. Later studies conducted on SINV revealed that nsPs colocalize with dsRNA at the neck of the spherules, strengthening the view that these membrane invaginations are the sites of alphavirus genome replication [112].

### 3.1. Alphavirus Replication Proteins

Spherule formation requires the assembly of functional replication complexes. Structural and functional studies revealed many of the functions of the individual nsPs that form the alphavirus replicase [49]. Among the four nsPs, nsP1 is the only protein that is capable of strongly associating with cellular membranes. The N-terminal domain of the ~60 kDa protein harbors a Rossmann-like methyltransferase motif that catalyzes the transfer of the methyl group from S-adenosylmethionine (AdoMet) to the N7 position of a GTP molecule (m^7^ GPPP). In addition to its MTase activity, nsP1 also possesses a guanylyltransferase (GTase) activity that allows for the transfer of the m^7^ GMP moiety to the viral RNA forming the cap-0 structure at the 5′ end [118,119,120]. The enzymatic activity of nsP1 is dependent on the binding of the protein to cellular membranes [119,121,122].

The 90 kDa nsP2 has multiple roles in the alphavirus life cycle [49]. Structural and functional studies conducted on nsP2 revealed that the protein has the following multiple domains: an N-terminal domain with helicase and nucleoside-triphosphatase (RTPase) activities, a central domain with papain-like protease activity, and a C-terminal domain harboring S-adenosyl methionine (SAM) methyltransferase-like activity [123,124]. The helicase/RTPase domain unwinds RNA secondary structures and removes a phosphate from the 5′ ends of nascent +RNAs, priming them for capping by nsP1 [125,126]. The protease domain cleaves the non-structural polyprotein p1234 into individual nsPs, while the methyltransferase-like domain plays a role in the cytopathic effect of the virus by inducing a host transcriptional shutoff [127,128,129,130].

The progress made in recent years in structural and functional studies has allowed for a better understanding of nsP3 and its role in the alphavirus life cycle [131,132]. The following three domains have been identified in nsP3: a macro domain, an alphavirus-unique (AUD) zinc-binding domain (ZBD), and a hypervariable phosphorylated domain (HVD) [133]. Studies conducted on CHIKV and VEEV showed that the macrodomain of nsP3 binds to monomeric ADP-ribose (MAR) and poly-ADP-ribose (PAR) [134]. This binding is implicated in counteracting the host immune response [135]. The central ZBD has been shown to be important for negative-sense and sgRNA synthesis and for the proper processing of the non-structural polyproteins [133,136]. Unlike the macro and zinc-binding domains, no structural information is available for the HVD of nsP3, and the sequence of this region is not well conserved among alphaviruses. This disordered domain functions as a hub for virus–host protein interactions, playing a major role in recruiting the host factors essential for the assembly of RCs [131,137]. The HVD of Old World and New World alphaviruses can interact with different components of stress granules—G3BP and FXR, respectively—an interaction that is necessary for the assembly of RCs as shown by CRISPR/Cas9-mediated knockout studies [138,139,140].

Among the four nsPs, the RNA-dependent RNA polymerase (RdRp) nsP4 is the most conserved [47,141]. NsP4 has an N-terminal domain that is unique for alphaviruses and important for replication, as well as a large C-terminal domain that harbors polymerase activity with features similar to other viral RdRps. NsP4 was also shown to have an adenylyltransferase (TATase) activity that is implicated in the polyadenylation of the alphavirus RNA [142,143]. Recently, the crystal structure of the RRV nsP4 was solved [144]. Based on the structure, nsP4 is highly dynamic and adopts a right-hand RdRp structure, forming finger, palm, and thumb domains. Interestingly, the structure also revealed that the alphavirus-unique N-terminal domain does not interact with the RdRp domain and is not required for the polymerase activity. However, in the presence of the N-terminal domain, RNA synthesis is enhanced, indicating that this domain might be a cofactor that enables the binding of the RdRp domain to RNA [144].

### 3.2. nsP1-Mediated Membrane Anchorage of the Replication Complex

Early studies conducted on the nsPs of SFV have shown that nsP1 is the only protein capable of tightly associating to cellular membranes [145]. This association was not disrupted by high salt or alkaline sodium carbonate treatments [145]. When expressed alone, nsP2 mainly localizes to the nucleus, nsP3 localizes to large aggregate-like cytoplasmic structures, and nsP4 localizes to the cytoplasm [146,147]. The flotation and immunostaining assays performed on cells expressing cleavable or uncleavable non-structural polyprotein intermediates have confirmed that only the uncleavable intermediates containing nsP1, such as P1234, P123, and P12, could localize to the plasma membrane, while P23 and P34 could not [146]. These observations validated the hypothesis that nsP1 is responsible for the membrane association of the alphavirus RC. NsP1 was later found to be capable of associating with negatively charged phospholipids such as PS via an amphipathic α-helical peptide (amino acids 245–264) located in the central portion of the protein [119,148]. Two mutations in SFV nsP1, R253E, and W259A, abrogated the membrane localization of the protein in mammalian cells and inhibited virus replication [148]. The palmitoylation of cysteine residues (418–420 in SFV) has been proposed to further strengthen the association of nsP1 with the plasma membrane [149]. Although not required for membrane association, the palmitoylation of nsP1 is important for the pathogenesis of SFV in vivo [150].

More recent studies conducted on nsP1 of CHIKV showed that the amphipathic α-helix of the protein is not sufficient for deforming liposomes and lipid nanotubes, indicating that other regions of the proteins are involved in membrane binding [151]. The exact mechanism of how nsP1 binds to membranes remained elusive until the structure of the protein from CHIKV was recently solved. The structure revealed that twelve copies of nsP1 assemble to form an 18.6-nanometer ring structure with a 7–7.5-nanometer-wide inner channel that allows for the transfer of RNA molecules and small globular proteins [121,122]. NsP1 rings were enzymatically active, whereas monomeric nsP1 was inactive, indicating that the oligomerization of the protein is required for its enzymatic activity. The upper portion of the ring structure harbors the MTase/GTase catalytic domain, while the lower portion is involved in anchoring the complex to the plasma membrane. Interestingly, both papers describing the structure of nsP1 found that two loops mediate membrane binding (loop 1: 200–235 and loop 2: 405–430). These two loops intertwine via hand-in-hand inter-loop bonding, forming membrane-binding spikes and bringing together neighboring nsP1 molecules. This interaction is crucial for activating the enzyme by stabilizing the two catalytic domains essential for its function [121,122]. Together, the new structures of nsP1 highlight the importance of its membrane association and confirm that nsP1 forms the base for the assembly of the RC at the neck of replication spherules (Figure 1). Future research should investigate the host factors that are recruited by nsP1 to the site of replication as well as their role in pathogenesis.

### 3.3. Viral and Host Factors Involved in the Formation of Replication Spherules

Despite the progress made over the past years in the field of alphavirus replication, the steps leading to the formation of replication spherules are still poorly understood. Studies conducted on SFV have shown that spherule formation depends on the presence of all four nsPs and is inhibited by the inactivation of any enzymatic function required for RNA synthesis [48]. The presence of an RNA template is also required for spherule biogenesis, as shown by using an SFV *trans*-replication system [152]. In contrast to spherules induced by viruses such as Flock House virus (FHV), those induced by SFV are dependent on the length of the RNA molecule, where longer RNA templates resulted in larger spherules [153,154]. In contrast, immunofluorescence and EM studies performed by the same group have shown that the spherules could form in the absence of an RNA template if P123 or P23 remain uncleaved in the presence of nsP4 [155]. However, membrane invaginations formed in the absence of a template were variable in size and significantly smaller than those formed in the presence of an RNA template [155].

Based on these findings, it is now evident that all four nsPs are required for the biogenesis of replication spherules. However, the essential host factors needed for this process are yet to be determined. Sequence analyses of SFV, CHIKV, and SINV have identified a proline-rich region in the nsP3 HVD, capable of high-affinity binding to the SH3 domain of amphiphysin-1 and Bin1/amphiphysin-2 [156,157]. A similar proline-rich domain has also been found in the NS5A protein of the HCV capable of binding to amphiphysin-SH3 [158]. Amphiphysins can drive membrane curvature via their highly conserved Bin/amphiphysin/Rsv (BAR) domain [159]. Interestingly, both mutating the amphiphysin-binding domain in nsP3 and knocking down amphiphysin in HeLa cells did not affect the formation of replication spherules but resulted in defects in RNA synthesis [156]. Therefore, the mechanism by which amphiphysin enhances alphavirus replication requires further studies. Since the new structure has shown that the ring complex formed by nsP1 can strongly associate with membranes through its rigid spikes, it is plausible that the membrane curvature is also induced by nsP1. The positively charged residues found on the edge of the midsection of the ring complex can also interact with the membrane phospholipid heads. Altogether, these interactions possibly induce the membrane bending necessary for spherule formation [121].

Live-cell imaging coupled with EM analyses have shown that replication spherules are internalized from the plasma membrane at a later stage during SFV infection [160]. The addition of Wortmannin, a phosphatidylinositol-3-kinase (PI3K) inhibitor, and blebbistatin, a myosin inhibitor, negatively affected the internalization of spherules from the plasma membrane (Figure. 1). On the other hand, a microtubule-polymerization inhibitor, Nocodazole, did not inhibit the internalization of spherules, instead resulting in their accumulation on small intracellular vesicles. Taken together, these results have shown that spherules are endocytosed from the plasma membrane in a PI3-kinase/actin-dependent manner, subsequently trafficking from small vesicles to acidic organelles via microtubules. Interestingly, the HVD of CHIKV and SFV nsP3 has been found to be the activator of the PI3K-Akt pathway [161]. A YXXM motif present in the HVD of RRV and SFV nsP3 has been implicated in binding to the PI3K regulatory subunit p85. This interaction liberates P110, which can then catalyze the conversion of Phosphatidylinositol 4,5-bisphosphate (PIP2) to Phosphatidylinositol (3,4,5)-trisphosphate (PIP3), resulting in the recruitment and activation of Akt (Figure 1) [162]. Research focusing on unraveling the wide virus–host interaction network would lead to a better understanding of the RCs and virus-induced structures associated with alphavirus replication.

## 4. Assembly Induced Membrane Rearrangements

The ER, *trans*-Golgi network (TGN), and CPV-IIs are the major intracellular membrane components that contain glycoproteins E1 and E2 [163,164,165]. It is proposed that CPV-IIs are derived from the TGN, first appearing in infected cells approximately four hours post-infection [69]. CPV-IIs have dimensions of approximately 100–400 nm by 1–2 µm and have NCs attached to them [6,70]. CPV-IIs can be comprised of single- or double-membranes; those that are single-membrane have NCs attached to their cytoplasmic face, while those that are double-membrane have NCs attached to both the inside and outside [166]. CPV-IIs may contain tubular formations of glycoprotein spikes arranged in hexagonal configurations that are approximate to their eventual organization on the mature viral envelope (Figure 2) [70]. Therefore, it is believed that CPV-IIs act as delivery vehicles, concentrating NCs and glycoproteins and transporting them via the secretory pathway to the plasma membrane for assembly and budding. This is in accordance with their tendency to be found close to the plasma membrane in the latest stages of infection [70]. More functional studies are needed, however, to confirm these speculations. Mutating the cdE2 endodomain has revealed that NCs associate with the outer surface of CPV-IIs through Cp-E2 interactions, in a similar fashion to the interactions that occur during virus budding [69,167,168].

Little is known regarding the host and viral factors that promote CPV-II biogenesis, but siRNA screening has revealed that ADP-ribosylation factor 1 (Arf1) may play a significant role [6]. Arf1 is a small GTPase that, upon activation, can regulate the formation of coat protein complex I (COPI) vesicles along the secretory pathway [169]. Arf1 can also modulate the actin cytoskeleton in a Cell Division Cycle 42 (CDC42), Neural Wiskott-Aldrich syndrome protein (N-WASP), and seven-subunit actin-related proteins-2/3 (Arp2/3) dependent manner, regulating the vesicular transport from the TGN to the plasma membrane [170,171,172]. Multiple viruses have been shown to interact extensively with Arf1 to control and restructure the Golgi apparatus. For example, the virus-induced manipulation of Arf1 by human immunity-related GTPase M (IRGM) in an HCV infection has been shown to induce Golgi fragmentation [173]. IRGM is known to contribute to autophagy; in response to HCV infection, it regulates the fragmentation of Golgi membranes by controlling the phosphorylation of Golgi-specific brefeldin A-resistance guanine nucleotide exchange factor 1 (GBF1). GBF1 is a guanine nucleotide exchange factor for Arf-GTPases; its manipulation by IRGM results in the altered functioning of Arf1 and the subsequent fragmentation of the Golgi. This results in the colocalization of Golgi vesicles with a replicating HCV [173]. Furthermore, the SARS-CoV ORF3a is both necessary and sufficient for coronavirus-induced Golgi fragmentation, but this can be inhibited with Arf1 overexpression [174]. The ORF3a protein in SARS-CoV is an ion channel; it disrupts Golgi morphology by neutralizing the pH and encouraging its reorganization [175]. It is hypothesized that Arf1 plays a similar role in alphavirus infection as it does in these other viruses, as Arf1 is the key regulator of the normal Golgi structure and integrity [176]. 

After originating from the TGN, it has been hypothesized that CPV-IIs traffic along actin filaments using a mechanism involving RAS-related C3 botulinum toxin substrate 1 (Rac1) (a downstream effector of Arf1), Arp3, and Phosphatidylinositol-4-Phosphate 5-Kinase Type 1 Alpha (PIP5K1-α), all regulators of actin polymerization and remodeling [6]. Actin-mediated trafficking has also been implicated in vaccinia virus dissemination [177]; specifically, vaccinia actin-based motility relies on Rac1 and its downstream effector Formin Homology 2 Domain Containing 1 (FHOD1), as well as the N-WASP/ARP2/3 pathway, possibly through a receptor tyrosine kinase-family mediator that integrates the two pathways [178]. An equivalent mediator may exist to play an analogous role during alphavirus infection, integrating these two pathways to coordinate actin nucleation and polymerization for CPV-II transport. Similar phenomena have been observed in Ebolavirus nucleocapsid transport [179] and intracellular baculovirus motility [180]. Future alphavirus research should look into the interplay between CPV-IIs, the cytoskeleton, and these known mechanisms of origination and trafficking.

## 5. Filopodial Extensions

Alphaviruses can modulate the plasma membrane by hijacking the host cytoskeleton and inducing the formation of short and long filopodial extensions [181,182]. Short extensions are induced by nsP1 via an unknown mechanism, as studies have shown that nsP1 expression alone induces the formation of short extensions that are nearly identical to those induced during an alphavirus infection [181]. Additionally, cells expressing only the structural proteins produce virus-like particles (VLPs), yet no short filopodial extensions [182]. Short filopodial extensions are ~2–7 μm in length and are mainly composed of actin, containing no microtubules (Figure 2) [183]. Though imaging data have revealed that SINV particles bud in large quantities from these extensions [166], the role of short filopodial extensions in assisting with assembly and budding is yet to be determined. This does, however, suggest a general purpose for these extensions as specialized assembly or budding sites. Further support for this model has been obtained using imaging studies, as most short extensions contain E2, and those that contain E2 also contain all of the other structural proteins [183]. Short filopodial extensions may also help to prevent the superinfection caused by the re-entry of newly formed virions back into the cell [184]. Interestingly, studies show that SFV, with a defect in filopodial extension formation due to a mutation that inhibits nsP1 palmitoylation, is attenuated in vivo [150]. This reveals that short filopodial extensions may also play a role in SFV pathogenesis, possibly by enhancing cell-to-cell spread, though more studies are required to confirm these speculations.

Long intercellular extensions are at least 10 μm in length, often reaching lengths of up to 60 μm [183]. They have been observed upon infection with several alphaviruses, including SINV, SFV, CHIKV, and VEEV, and have been observed in several different cell types, including mammalian fibroblasts, mammalian epithelial cells, and mosquito cells [166,182,183]. Their formation is not well understood, but is promoted by the alphavirus structural proteins; in particular, they require the presence of E2 at the cell surface and its interaction with Cp [6,182]. In addition to E2 and Cp, E1 can be found throughout the length of the extension [6]. This type of filopodial extension is microtubule positive and actin positive, and they promote contact between infected and uninfected cells [182,183]. Hence, these long extensions can possibly promote cell-to-cell transmission [182,185]. Long filopodial extensions do not fuse with the target cell; in other words, though stable physical contacts between the two cells form, the cytoplasm of the infected cell and the target cell do not ever exchange [183]. Furthermore, long filopodial extensions have only ever been observed emanating from infected cells, and interestingly, cell-to-cell transmission mediated by long filopodial extensions is relatively insensitive to receptor downregulation and neutralizing antibodies [182]. Therefore, long filopodial extensions may assist in delivering newly formed virions to nearby cells in an effort to evade the host immune system [185]. Future studies focusing on the host factors involved in the formation of alphavirus-induced filopodial extensions would help achieve a better understanding of the role of these structures in alphavirus spreading.

## 6. Alphavirus-Induced Structural Modifications in Mosquito Cells

Though alphaviruses establish cytolytic infections in mammalian cells, they cause noncytopathic, persistent infections in mosquito cells [186]. Their ability to persevere in mosquitoes, which interact intimately with humans in warm and often urban areas, contributes greatly toward the transmission of epidemic strains of alphaviruses. Recent studies have allowed for the characterization of alphavirus-induced membrane rearrangement in mosquito cells. The analysis of replication and assembly in two distinct hosts will hopefully pave the way for understanding the mechanisms of alphavirus maintenance in nature.

The replication and growth kinetic analyses of baby hamster kidney (BHK) and *A. albopictus* clone C6/36 cells have revealed that SINV replication occurs at a lower rate in mosquito compared to mammalian cells [166]. Furthermore, in BHK cells, alphavirus RCs are distributed throughout the cytoplasm, at internal vesicles, and on the plasma membrane. This is different from C6/36 cells, where RCs are distributed primarily on the outer membrane of large cytopathic vacuoles containing E2, not on the plasma membrane [166]. Most notably, though CPV-I and CPV-II structures are easily visible using transmission EM in BHK cells, infected mosquito cells lack the classical CPV-I and CPV-II. Rather, throughout the infection, there are large cytopathic vacuoles with properties intermediate to CPV-I and CPV-II [166]. These structures contain replication spherules, similar to classical CPV-I, and are studded with NCs on the outside and viral glycoproteins on the inside, similar to classical CPV-II. These vacuoles also contain internally budded virus particles that are eventually secreted as individual virions. In persistently infected C6/36 cells, the E2 glycoprotein associates in large quantities with large cytopathic vacuole membranes; meanwhile, virus replication and particle production are significantly reduced, yet steady [166]. Persistently infected C6/36 cells were also shown to directly transport budded SINV virions to uninfected cells using long filopodial extensions, bypassing the extracellular medium [166].

Taken together, the strategy for virus replication in mosquito cells is not as highly compartmentalized as in mammalian cells. Rather, mosquito cells employ tactics of replication and assembly that intermingle by means of a large cytopathic vacuole intermediate to CPV-I and CPV-II. Additionally, alphaviruses employ internal budding in mosquito cells in addition to classical budding from the plasma membrane, and are capable of direct cell-to-cell transmission. These strategies appear key for transitioning from acute to persistent infection, enabling alphaviruses to thrive in nature. More work is needed, however, to explore how and why these alternative membrane rearrangements and transmission tactics help alphaviruses establish persistent infection.

## 7. Conclusions

Though significant strides in alphavirus research have been made in recent years, there are still several outstanding questions that remain regarding how and why alphaviruses induce structural modifications in their hosts. What host factors are involved in replication spherule and cytopathic vacuole biogenesis? What are the minimum requirements for CPV-II formation? How does nsP1 induce short filopodial extension formation? Why do mosquito cells undergo alternative structural modifications upon alphavirus infection, and how does this help them reach a state of persistent infection? Live imaging techniques or small molecule inhibitors should be used to elucidate the functions of all alphavirus proteins. Resolving the cryo-EM structure of the alphavirus replication complex will also help shed light on these outstanding questions. These studies will help reach the ultimate, overarching goal of alphavirus research: to discover novel antivirals and therapeutics that will stop the spread of alphavirus infections before they become more of a widespread problem than they already are.

## Figures and Tables

**Figure 1 pathogens-10-00984-f001:**
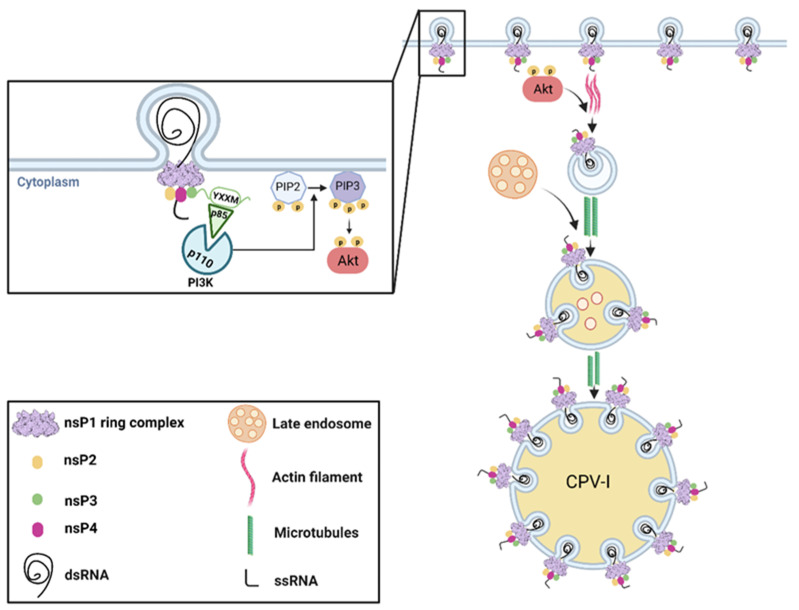
Alphavirus-induced replication organelles. Alphavirus replication spherules form at the plasma membrane and require the presence of nsP1, nsP2, nsP3, and nsP4. Structural studies proposed that the nsP1 ring complex forms the base of the alphavirus RC and plays a role in membrane curvature. The HVD of nsP3 contains a YXXM motif capable of binding to p85, relieving p110 inhibition. P110 catalyzes the phosphorylation of PIP2, generating PIP3, which can recruit and activate Akt at the plasma membrane. Active Akt promotes spherule internalization from the plasma membrane in an actin-myosin-dependent manner. Endocytic spherule-containing vesicles fuse with late endosomes to form acidic vesicles that traffic to the perinuclear area via microtubules, where they mature to form large CPV-Is. Figure created with BioRender.com.

**Figure 2 pathogens-10-00984-f002:**
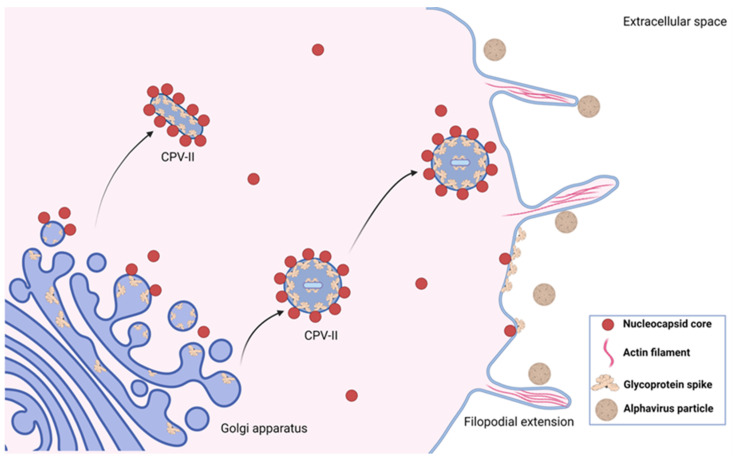
Structures formed during alphavirus assembly. CPV-IIs originate with variable morphologies from the *trans*-Golgi network. These structures have multiple NCs associated with them, and some contain tubular formations of glycoprotein spikes. CPV-IIs are presumably trafficked to the plasma membrane to deliver NCs and mature glycoproteins. Alphavirus infection also results in actin-rich filopodial extensions that facilitate budding and cell-to-cell spread. Figure created with BioRender.com.

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
