# Peer review of "Alphavirus-Induced Membrane Rearrangements during Replication, Assembly, and Budding"

_pathogens, 2021, doi:10.3390/pathogens10080984_

Round 1

Reviewer 1 Report

The manuscript entitled "Alphavirus-Induced Structural Modifications During Replication, Assembly, and Budding" elegantly summarizes the current knowledge of alphavirus-induced structures and their functions in infected cells, focusing on both replication-induced and assembly-induced membrane rearrangements. The work brings together important pieces of information from the alphavirus literature, including those related to structural modifications induced in mosquito cells. However, three minor issues - described below in order of appearance in the manuscript - needs to be addressed by the authors:

- Line 34: Despite being an arthritogenic alphavirus of the Semliki Forest complex like CHIKV, MAYV is a New World alphavirus according to its area of origin.
- Lines 94-95: Some alphaviruses species may transiently retain E3 protein as a structural component noncovalently attached to the spike trimers.
- Lines 108-109: Previous studies using SFV (DOI: 10.1128/JVI.79.12.7942-7948.2005) and MAYV (DOI: 10.1021/acs.biochem.7b00650) have indicated that low-pH induced conformational changes in alphavirus E1/E2 heterodimers may be at least partially reversible.

Author Response

Comments of Reviewer #1, General

The manuscript entitled "Alphavirus-Induced Structural Modifications During Replication, Assembly, and Budding" elegantly summarizes the current knowledge of alphavirus-induced structures and their functions in infected cells, focusing on both replication-induced and assembly-induced membrane rearrangements. The work brings together important pieces of information from the alphavirus literature, including those related to structural modifications induced in mosquito cells. However, three minor issues - described below in order of appearance in the manuscript - needs to be addressed by the authors:

We thank the reviewer for the helpful comments. We have incorporated changes accordingly.

- Line 34: Despite being an arthritogenic alphavirus of the Semliki Forest complex like CHIKV, MAYV is a New World alphavirus according to its area of origin.

Resolved. Added the following sentence with a new reference: An exception to encephalitic New World is Mayaro virus (MAYV) which is arthritogenic.

- Lines 94-95: Some alphaviruses species may transiently retain E3 protein as a structural component noncovalently attached to the spike trimers.

We have added the following sentence with two references: Alphaviruses such as SFV and VEEV have been shown to transiently retain E3 in association with E2 after furin cleavage in the mature virion.

- Lines 108-109: Previous studies using SFV (DOI: 10.1128/JVI.79.12.7942-7948.2005) and MAYV (DOI: 10.1021/acs.biochem.7b00650) have indicated that low-pH induced conformational changes in alphavirus E1/E2 heterodimers may be at least partially reversible.

We have rewritten the to include the suggested changes.

Modified text: causing a conformational change in the E1/E2 heterodimer, triggering the formation of fusion pores in both the cellular and viral membrane to allow for release of the NC into the cytoplasm [4,42,43]. However, studies on SFV and MAYV showed that the E1/E2 conformational changes can be partially reversed upon reneutralization of the acidic PH [44,45].  Low-PH induced conformational changes of the glycoprotein spikes occurs rapidly once sufficient acidification is reached in order to avoid lysosomal degradation [39].

Reviewer 2 Report

The authors provide here a clear, complete and very interesting review focused on the changes in composition and organization of cellular compartments which occur during alphavirus infection, in mammal and mosquito cells.

After a short introduction on Alphaviruses taxonomy and structure, they give a very detailed overview of alphavirus cycle. The major part of the review is then focused on the membrane rearrangements induced by viral replication, detailing the implication of viral or mammalian host factors. After an overview of assembly and budding-induced cellular modifications, they detail the actual (and limited) knowledge on alphavirus-induced cell structure modifications in mosquito cells.

Comments :

- the title is a bit confusing with the term “structural modifications”, which may not be linked to cellular modifications at first sight. The authors should find a title including the cellular aspect of these modifications.

- if possible, a scheme of the viral cycle, in mammal and mosquito cells, which summarize the different models described could be helpful for the “take-home” message of the review.

- line 81-84 : in the Rose et al. article, NRAMP2 is genetically downregulated only in MEF but mammalian, drosophila and mosquito cells are treated with Iron, with the consequent NRAMP downregulation and other effects, especially in cellular trafficking and membrane structures.

Reference n°24 is linked to a High-Affinity Laminin Receptor and not NRAMP2.

- line 256 : the subchapter 3.2 title is not in the right font.

- some abbreviations are not described in the text : TIM (line 72); PIP2/PIP3 (line 347); GBF1 (line 375); PIP5K1-a, FHOD1 and NWASP (line 389/390)

- Figure 1 : in the legend, p85 and p110 are not described as PI3K subunits and “PI3K” is really hard to read on the scheme. It would be clearer if PI3K was written bigger under the blue circle.

AKT is written in capitals in the figure and legend but “Akt” in the text (line 348)

The red circles in the spherule after late endosome fusion are not described.

- line 318 : “that” is repeated

- line 370 : Arf implication in membrane trafficking and modification should be briefly explained.

- line 443 : “spreading” would be more appropriate than “pathogenesis”.

- line 470 : the reference n°179 seems to be more related to line 439, in the paragraph concerning mammals.

“This phenomenon has also been observed in mammalian cells infected with CHIKV” : this sentence seems unnecessary, as it is described just above.  

- It is now known that 3D or polarized culture versus monolayer cell culture can induce strong changes in cell metabolism, trafficking and response to stress, such as viral infection. Could these alphavirus-induced intracellular structures (CPV-I, II and especially filopodial extensions) be affected by a 3D or 2D environment?

Author Response

Comments of Reviewer #2, General:

The authors provide here a clear, complete and very interesting review focused on the changes in composition and organization of cellular compartments which occur during alphavirus infection, in mammal and mosquito cells.

After a short introduction on Alphaviruses taxonomy and structure, they give a very detailed overview of alphavirus cycle. The major part of the review is then focused on the membrane rearrangements induced by viral replication, detailing the implication of viral or mammalian host factors. After an overview of assembly and budding-induced cellular modifications, they detail the actual (and limited) knowledge on alphavirus-induced cell structure modifications in mosquito cells.

We thank the reviewer for the critical analysis, feedback and questions.

Comments :

- the title is a bit confusing with the term “structural modifications”, which may not be linked to cellular modifications at first sight. The authors should find a title including the cellular aspect of these modifications.

Based on this comment, we have changed the title.

New title: Alphavirus-Induced membrane rearrangements During Replication, Assembly, and Budding

- if possible, a scheme of the viral cycle, in mammal and mosquito cells, which summarize the different models described could be helpful for the “take-home” message of the review.

We thank the reviewer for this suggestion. Although it is helpful to see the different life cycles in different hosts, we have published a detailed illustration of the alphavirus life-cycle in mammalian and mosquito cells earlier in the following paper: DOI: 10.2217/fmb.09.59. Since the focus of this review is membrane modifications in infected cells, we prefer the figures to be focused on this topic. We, therefore, decided not to make any new life cycle figures for this review.

- line 81-84 : in the Rose et al. article, NRAMP2 is genetically downregulated only in MEF but mammalian, drosophila and mosquito cells are treated with Iron, with the consequent NRAMP downregulation and other effects, especially in cellular trafficking and membrane structures.

We thank the reviewer for the feedback on the unfortunate confusion caused by our statement. The text has been rewritten to include the suggested changes.

Modified text: Natural resistance-associated macrophage protein (NRAMP) has been proposed as a potential entry receptor for SINV, as the genetic downregulation of NRAMP2 in mouse fibroblasts, and the post-translational downregulation of NRAMP by iron treatment in insect cells all resulted in reduced SINV infection

Reference n°24 is linked to a High-Affinity Laminin Receptor and not NRAMP2.

We have removed the reference.

- line 256 : the subchapter 3.2 title is not in the right font.

We thank the reviewer for noticing this. This issue has been resolved.

- some abbreviations are not described in the text : TIM (line 72); PIP2/PIP3 (line 347); GBF1 (line 375); PIP5K1-a, FHOD1 and NWASP (line 389/390)

We thank the reviewer for this helpful comment. We have added the abbreviations as highlighted in the main text.  

- Figure 1 : in the legend, p85 and p110 are not described as PI3K subunits and “PI3K” is really hard to read on the scheme. It would be clearer if PI3K was written bigger under the blue circle.

We thank the reviewer for this observation. We have modified the figure accordingly.

AKT is written in capitals in the figure and legend but “Akt” in the text (line 348)

We have changed the text.

The red circles in the spherule after late endosome fusion are not described.

The color was changed to match the vesicles present in late endososmes. The fusion of spherule-containing vesicles with the late endosome results in the presence of these circles/vesicles.

- line 318 : “that” is repeated

We have fixed the typo.

- line 370 : Arf implication in membrane trafficking and modification should be briefly explained.

We thank the reviewer for this suggestion. We expanded on the role of Arf1 and added the following sentences: Arf1 is a small GTPase, that upon activation can regulate the formation of coat protein complex I (COPI) vesicles along the secretory pathway. Arf1 can also modulate the actin cytoskeleton in a CDC42, N-WASP, and Arp2/3 dependent manner regulating vesicular transport from the TGN to the plasma membrane.  

- line 443 : “spreading” would be more appropriate than “pathogenesis”.

We have used the suggested word.

- line 470 : the reference n°179 seems to be more related to line 439, in the paragraph concerning mammals.

“This phenomenon has also been observed in mammalian cells infected with CHIKV” : this sentence seems unnecessary, as it is described just above.  

We thank the reviewer for noticing this. We moved the reference to the suggested section and deleted the unnecessary sentence.

- It is now known that 3D or polarized culture versus monolayer cell culture can induce strong changes in cell metabolism, trafficking and response to stress, such as viral infection. Could these alphavirus-induced intracellular structures (CPV-I, II and especially filopodial extensions) be affected by a 3D or 2D environment?

We thank the reviewer for this insightful comment. Although this is an interesting observation worth pursuing, and there are publications of CHIKV infected polarized cells, we could not locate literature specifically showing the ultrastructure of polarized cells infected with alphaviruses. Since those changes have not been published, we decided not to include aspects concerning polarized cells in this review.